# The Cytotoxic Properties of Some Tricyclic 1,3-Dithiolium Flavonoids

**DOI:** 10.3390/molecules24132459

**Published:** 2019-07-04

**Authors:** Laura G. Sarbu, Sergiu Shova, Dragos Peptanariu, Isabela A. Sandu, Lucian M. Birsa, Lucian G. Bahrin

**Affiliations:** 1Alexandru Ioan Cuza University of Iasi, Department of Chemistry, 11 Carol I Blvd., 700506 Iasi, Romania; 2Petru Poni Institute of Macromolecular Chemistry, Intelcenter. 41A Grigore Ghica Vodă Alley, 700487 Iasi, Romania

**Keywords:** flavonoids, dithiolium, cytotoxicity

## Abstract

Background: Due to the emergence of multidrug resistant microorganisms, new classes of antibiotics are needed. In this paper, we present the cytotoxic effects of five tricyclic flavonoids, one of which was previously identified as a potent antimicrobial agent. Methods: All five derivatives were tested against human HOS and MCF7 cancer cell lines using a wound scratch assay. The cytotoxic properties of previously reported flavonoid **4a** were also evaluated using the standard MTS (3-(4,5-dimethylthiazol-2-yl)-5-(3-carboxymethoxyphenyl)-2-(4-sulfophenyl)-2*H*-tetrazolium, inner salt) and live/dead assays, using NHDF, HOS and MCF7 cell lines. Results: All five derivatives were found to inhibit to some degree the proliferation of cancer cells. **4a** was also found to be less toxic towards regular versus cancerous human cells. Moreover, the minimum bactericidal concentration of **4a** against *Staphylococcus aureus* was found to be non-toxic for any of the tested human cell lines. Conclusions: Derivative **4a** has the potential of being used as a therapeutic agent against certain microorganisms. Further structure optimization is required for use against tumors.

## 1. Introduction

Since the discovery of penicillin in 1928 and until present day, antibiotics have been used extensively, for both medical and agricultural purposes. This ultimately led to the emergence of multidrug resistant microorganisms, as pathogens acquired resistance to multiple classes of antibiotics [1]. As such, the need to develop new drugs is now greater than ever [2]. Moreover, it is desirable that said drugs belong to novel classes of antibiotics, to which microorganisms have not had time to adapt to [3]. In this respect, flavonoids are good candidates when looking for new antimicrobial agents. As natural polyphenols, flavonoids are widely distributed in the plant kingdom and present a variety of biological activities, such as antiviral [4], antifungal [5], anti-inflammatory [6], antitumoral [7], cardio-protective [8], neuro-protective [9] and antibacterial [10] properties.

With respect to their antibacterial properties, the most studied flavonoids are the naturally occurring ones. However, it was shown that semi-synthetic flavonoids, obtained by modifying natural ones, can lead to better antimicrobial agents [11,12]. Moreover, several synthetic flavonoids were also found to display potent antimicrobial properties, a subject which was recently reviewed by our group [13].

Although strong antimicrobial properties are desired, they are not enough when looking for new antibiotics. Thus, any new drug must also be well tolerated by the human body, inflicting as little damage as possible to human cells. Therefore, once the antimicrobial capabilities of a new agent have been established, cytotoxicity studies are needed.

Our continued interest in synthetic flavonoids has prompted us to synthesize and evaluated the cytotoxic effect of five tricyclic flavonoids, whose backbone is known to induce antimicrobial properties [14,15,16,17,18]. Out of the five, three are reported here for the first time.

## 2. Results and Discussion

### 2.1. Chemistry

The synthetic route used to obtain 1,3-dithiolium flavonoids **4a**–**e** is described in Scheme 1 and follows the protocol used for the model compound **4a** [19].

Methylated flavonoids **4b**–**d** were thought as an extension to our previous structure-activity studies [16,17], which found that the optimal structure when it comes to antibacterial properties contains either a Br or I substituent on the benzopyran core and a Cl or Br substituent on the side-ring, while unsubstituted derivatives at the benzopyran ring and 2-aryl moiety displayed the lowest antibacterial acitivity.

The first step of the synthesis sees the formation of the benzopyran core, by reacting previously reported phenacyl dithiocarbamates **1a**–**e** with aminal **2**, leading to dithiocarbamic flavanones **3a**–**e**. This kind of reaction, used by our group in the past [20], runs smoothly in refluxing ethanol, over a period of 2 h. The reaction product, which precipitates upon cooling, is filtered and recrystallised from ethanol. The formation of flavanones **3a**–**e** is confirmed by NMR and mass spectrometry. ^1^H NMR analysis indicates the disaprearance of the phenolic protons of **1a**–**e**, present around 11.00–12.00 ppm. In turn, two new doublets appear between 5.00–7.00 ppm, corresponding to the H-2 and H-3 protons of the newly formed benzopyran core. Because these two protons can be located either on the same side or on opposite sides of the plane of the molecule, two stereomers, namely *syn*-**3** and *anti*-**3** can be obtained. For all new dithiocarbamic flavonoids **3b**–**e** the *syn/anti* ratio was always identified to be 4:6. This was established on the basis of coupling constants between the H-2 and H-3 protons. The *anti* isomers always displayed a coupling constant between 9.5–10.7 Hz and the *syn* isomers around 2.8 Hz. ^13^C NMR analysis confirms the presence of the C-2 carbon atom, found around 80.0 ppm, while the C-3 carbon atom can be found around 60.0 ppm.

The next step, the closure of the 1,3-dithiolium ring, was performed by heating flavanones **3a**–**e** in a mixture of acetic acid and sulfuric acid, followed by treatment with an aqueous solution of sodium tetrafluoroborate. The desired products, obtained as colorless precipitates, are filtered and recrystallised from ethanol. ^1^H NMR analysis indicates the disappearance of the H-3 signal and a shift of the H-2 signal to about 6.80 ppm, while the ^13^C NMR spectra show the disappearance of the carbonyl and thiocarbonyl signals from around 190.0 ppm. Instead, a new signal belonging to the positive carbon atom of the 1,3-dithiolium ring can be found around 185.0 ppm.

### 2.2. X-Ray Crystallography

The structure of tricyclic flavonoid **4b** has been unambiguously proved by single crystal X-ray diffraction study [21]. It was confirmed that compound **4b** crystalizes in the *P*2_1_/n space group of monoclinic system. Its ionic crystal structure consists of flavonoid cations and BF_4_^-^ anions in 1:1 ratio. No co-crystallized solvent has been found in the crystal. The result of X-ray diffraction investigation is shown in Figure 1, while the bond lengths and angles are summarized in Table 1. It is to note, that the chlorophenyl fragment, as well as one of the sulphur atoms were found to be disordered into two resolvable positions with components ratio refined at 0.7:0.3.

The bond length C10–N1 of 1.311(2) Å corresponds to a double bond and thus confirms the formal positive charge at the nitrogen atom, while C3–C4 is of 1.342(2) Å indicating a double bond character. The C–S bond distances in the five-membered ring vary in the range between 1.727(1)–1.746(1) Å. The crystal structure of compound **4b** essentially results from the parallel packing of the discrete two-dimensional supramolecular layers (Figure 2), where the cationic and anionic components are interacting through C-H---F hydrogen bonding. The further analysis of the 2D architecture shows the presence of π–π stacking between bromophenyl rings of two centrosymmetrically related cations, which is evidenced by centroid-to-centroid distance of 3.700(8) Å.

### 2.3. Wound Scratch Assay

Wound scratch assay is commonly used to evaluate migration and proliferation of tumor cells in response to a treatment. We wanted to know if our compounds have any antitumor activity, therefore their effect was investigated on two neoplastic cell lines: HOS-osteosarcoma and MCF7-a breast cancer cell line (Figure 3; Appendix A).

The test consists of producing a wound on cell culture and tracking the evolution of the open area created while cells were under the influence of drugs to be tested. All compounds demonstrated inhibition on cell proliferation and migration on both cell lines; however the most effective ones proved to be **4a**, **4b** and **4c**. Upon treatment of HOS cells with these three compounds, the wound was between 5.6 to 6.6 times slower to heal than control, while in the case of MCF7 cells the inhibition of wound healing after treatment with **4a**, **4b** and **4c** was around 0.3.

The inhibition induced by **4d** and **4e** relative to control on HOS cells was about 2.4 and 3.3 fold change respectively, while on MCF7 cells they were 0.24 and 0.20 respectively.

Encouraged by these results, we decided to further investigate the cytotoxic effects of flavonoids of type **4** on normal and cancerous human cells. Considering that the antibacterial properties of **4a** were found by us to be very promising (*S. aureus*—MIC and MBC = 0.24 µg/mL; *E. coli*—MIC = 3.9 µg/mL, MBC = 7.8 µg/mL) [15], we decided to focus on it as a model compound.

### 2.4. MTS Assay

A metabolic test CellTiter 96^®^ AQueous One Solution Cell Proliferation Assay (MTS) which measures the mitochondrial reductase activity was used to investigate the influence of **4a** on cells. Cells with unaffected mitochondrial function could reduce the MTS reagent to a soluble and colorful formazan and as a result, the mitochondrial reductase activity is used as an indirect indicator for cell viability.

After 48 h of treatment, **4a** (Figure 4) demonstrates greater cytotoxic activity on HOS versus NHDF cells at concentrations between 1.7–8 µg/mL, statistically significant in the interval 2.6–4.1 µg/mL. Likewise, **4a** is killing MCF7 cells more efficiently than NHDF cells in the interval 3.3–6.4 µg/mL. In terms of magnitude of effect the interval 4.1–5.1 µg/mL appears to be the most beneficial. Thus, the relative viability at 4.1 µg/mL **4a** for NHDF, HOS and MCF7 cells is 62%, 49% and 39% respectively, while at 5.1 µg/mL it is 62%, 49% and 31% respectively. At concentrations lower than 1.7 µg/mL for HOS and 3.3 µg/mL for MCF7, **4a** does not kill these tumoral cells better than fibroblasts, and at concentrations lower than 1.0 µg/mL all tested cells are virtually unaffected.

These results suggest that **4a** could potentially be used as an antitumor agent, although the rather high toxicity exhibited against normal cells is a drawback. Addressing this issue will require further research, in order to identify the optimal structure of compounds of type **4** that would decrease their effect on normal cells, while still retaining the ability to act against cancerous cells.

In a previous study, we also documented the antibacterial properties of **4a** [15]. It was then found that the minimum inhibitory concentration, as well as minimum bactericidal concentration against *S. aureus* is 0.24 µg/mL. At this concentration, the tested human fibroblasts are virtually unaffected, suggesting that tricyclic flavonoids similar to **4a** could be used as antimicrobial drugs against gram-positive bacteria.

### 2.5. Live/Dead Staining

MTS assay is a metabolic test which evaluates cell viability in an indirect manner. To get more insight over antitumor activity and cytotoxicity, a live/dead staining was performed, which offers more direct information.

The inhibition of proliferation and migration of malignant cells while maintaining a lower cytotoxic effect on healthy cells is a desirable feature of antitumor drugs. As could be seen in Figure 5, compound **4a** at a concentration of 5 µg/mL is visibly affecting the viability and proliferation of HOS and MCF7 cells (48 h control—untreated cells vs. 48 h treatment—with **4a**). These effects are noticeable for NHDF cells as well but to a lower extent, compared to the two cancerous cell lines.

These results confirm those recorded using the MTS assay, namely that **4a** affects the HOS and MCF7 cell lines to a greater extent than the NHDF cell line.

## 3. Experimental

### 3.1. Chemistry

Melting points were obtained on a *KSPI* melting-point meter and are uncorrected. Melting points of flavonoids **3b**–**d** concern the mixture of isomers. IR spectra were recorded on a Bruker Tensor 27 instrument. NMR spectra were recorded on a Bruker 500 MHz spectrometer. Chemical shifts are reported in ppm downfield from TMS. Mass spectra were recorded on a Thermo Scientific ISQ LT instrument. All reagents were commercially available and used without further purification.

#### 3.1.1. General Procedure for 8-bromo-2-(4-chlorophenyl)-6-methyl-4-oxochroman-3-yl *N*,*N*-diethyldithiocarbamate (**3b**)

To a solution of 1-(3-bromo-2-hydroxy-5-methylphenyl)-1-oxoethan-2-yl-*N*,*N*-diethyldithiocarbamate (**1b**) [22] (0.75 g, 2 mmol) in EtOH (30 mL) aminal **2** (0.59 g, 2 mmol) was added and the reaction mixture was heated under reflux for 2 h. After cooling the solid material was filtered off and purified by recrystallization from ethanol to give **3b** (0.82 g, 83%) as colorless crystals. M.p. 149–150 °C. IR (ATR, cm^−1^) 2889, 1692, 1492, 1271, 1239, 803, 649, 548. ^1^H NMR (DMSO-*d*6, selected data for the major isomer) δ 7.81 (d, *J* = 1.4 Hz, 1H), 7.62 (d, *J* = 1.4 Hz, 1H), 7.50 (m, 4H), 6.17 (d, *J* = 10.4 Hz, 1H), 5.85 (d, *J* = 10.4 Hz, 1H), 3.83 (m, 4H), 2.31 (s, 3H), 1.10 (m, 6H). ^13^C NMR (DMSO-*d*6, selected data for the major isomer) δ 189.8, 187.1, 154.8, 140.7, 135.0, 134.0, 133.3, 130.1, 128.8, 128.5, 126.8, 111.2, 80.0, 57.7, 50.1, 47.6, 20.1, 12.7, 11.5. MS (EI) *m*/*z*: 497.1 (M^+^, 7%) for C_21_H_21_^79^BrClNO_2_S_2_.

#### 3.1.2. 6-Bromo-2-(4-chlorophenyl)-8-methyl-4-oxochroman-3-yl *N*,*N*-diethyldithiocarbamate (**3c**)

The compound was synthesized from 1-(5-bromo-2-hydroxy-3-methylphenyl)-1-oxoethan-2-yl-*N*,*N*-diethyldithiocarbamate (**1c**) [23] following the general procedure. Colorless crystals (0.46 g, 80%). M.p. 178–179 °C. IR (ATR, cm^−1^) 2877, 1696, 1490, 1419, 1260, 1200, 811, 652, 539. ^1^H NMR (DMSO-*d*6, selected data for the major isomer) δ 7.70 (m, 2H), 7.50 (m, 4H), 6.13 (d, *J* = 10.7 Hz, 1H), 5.85 (d, *J* = 10.4 Hz, 1H), 3.82 (m, 4H), 2.19 (s, 3H), 1.10 (m, 6H). ^13^C NMR (DMSO-*d*6, selected data for the major isomer) δ 189.9, 186.9, 157.9, 139.6, 135.1, 133.3, 130.8, 130.1, 128.7, 126.8, 121.2, 113.7, 79.5, 57.7, 51.0, 47.6, 15.5, 12.9, 11.5. MS (EI) *m*/*z*: 496.0 (M^+^, 4%) for C_21_H_21_^79^BrClNO_2_S_2_.

#### 3.1.3. 6,8-Dibromo-2-(4-chlorophenyl)-7-methyl-4-oxochroman-3-yl *N*,*N*-diethyldithiocarbamate (**3d**)

The compound was synthesized from 1-(3,5-dibromo-2-hydroxy-4-methylphenyl)-1-oxoethan-2-yl-*N*,*N*-diethyldithiocarbamate (**1d**) [24] following the general procedure. Colorless crystals (0.83 g, 72%). M.p. 164–165 °C. IR (ATR, cm^−1^) 2888, 1697, 1490, 1417, 1337, 1199, 1011, 802, 642, 528. ^1^H NMR (DMSO-*d*6, selected data for the major isomer) δ 7.94 (m, 2H), 7.52 (m, 4H), 6.22 (d, *J* = 9.5 Hz, 1H), 5.89 (d, *J* = 9.5 Hz, 1H), 3.80 (m, 4H), 2.58 (s, 3H), 1.08 (m, 6H). ^13^C NMR (DMSO-*d*6, selected data for the major isomer) δ 190.7, 186.0, 158.3, 145.7, 135.5, 134.2, 130.3, 128.8, 128.7, 121.1, 117.1, 115.2, 82.5, 57.3, 50.7, 47.7, 24.9, 12.7, 11.4. MS (EI) *m*/*z*: 574.9 (M^+^, 10%) for C_21_H_20_^79^Br_2_ClNO_2_S_2_.

#### 3.1.4. 2-(4-Chlorophenyl)-6,8-diiodo-4-oxochroman-3-yl *N*,*N*-diethyldithiocarbamate (**3e**)

The compound was synthesized from 1-(3,5-diiodo-2-hydroxyphenyl)-1-oxoethan-2-yl-*N*,*N*-diethyldithiocarbamate (**1e**) [25] following the general procedure. Colorless crystals (0.92 g, 70%). M.p. 160–161 °C. IR (ATR, cm^−1^) 3078, 1685, 1570, 1491, 1422, 1261, 1200, 1085, 972, 819, 655, 513. ^1^H NMR (DMSO-*d*6, selected data for the major isomer) δ 8.39 (m, 1H), 8.03 (m, 1H), 7.51 (m, 4H), 6.20 (d, *J* = 10.7 Hz, 1H), 5.85 (d, *J* = 10.7 Hz, 1H), 3.81 (m, 4H), 1.08 (m, 6H). ^13^C NMR (DMSO-*d*6, selected data for the major isomer) δ 189.6, 186.3, 159.1, 152.2, 135.6, 134.6, 134.2, 133.5, 130.2, 128.7, 80.2, 57.2, 50.7, 47.7, 12.8, 11.5. MS (EI) *m*/*z*: 656.9 (M^+^, 26%) for C_20_H_18_ClI_2_NO_2_S_2_.

#### 3.1.5. General Procedure for 2-*N*,*N*-diethylamino-6-bromo-4-(4-chlorophenyl)-8-methyl-4*H*-1,3-dithiol [4,5-*c*]chromen-2-ylium tetrafluoroborate (**4b**)

To a mixture of sulfuric acid (0.5 mL) and acetic acid (1.5 mL), flavanone **3b** (0.5 g, 1 mmol) was added and the resulting solution was heated to 80 °C for 20 min. The reaction mixture was then left to cool to room temperature and a solution of sodium tetrafluoroborate (250 mg) in water (10 mL) was added dropwise, with vigorous stirring. The resulting precipitate was then filtered, washed thoroughly with water and recrystallized from ethanol, yielding the desired tetrafluoroborate **4b** in the form of colorless crystals (0.5 g, 88%). M.p. 232–233 °C. IR (ATR, cm^−1^) 1538, 1441, 1235, 1041, 774, 734, 460, 451. ^1^H NMR (DMSO-*d*6) δ 7.52 (m, 4H), 7.48 (d, *J* = 1 Hz, 1H), 7.31 (d, *J* = 1 Hz,) 6.92 (s, 1H), 3.94 (m, 4H), 2.27 (s, 3H), 1.41 (t, *J* = 7.1 Hz, 3H), 1.34 (t, *J* = 7.1 Hz, 3H). ^13^C NMR (DMSO-*d*6) δ 185.1, 145.7, 135.9, 135.6, 135.0, 134.6, 129.7, 129.6, 128.3, 127.9, 125.1, 117.9, 110.8, 75.1, 54.7, 54.6, 20.1, 10.8, 10.6. MS (EI) *m*/*z*: 480.1 (M^+^-BF_4,_ 7%) for C_21_H_20_^79^BrClNOS_2_]^+^.

#### 3.1.6. 2-*N*,*N*-Diethylamino-8-bromo-4-(4-chlorophenyl)-6-methyl-4*H*-1,3-dithiol[4,5-*c*]chromen-2-ylium tetrafluoroborate (**4c**)

Colorless crystals (0.48 g, 84%). M.p. 225–226 °C. IR (ATR, cm^−1^) 1549, 1454, 1210, 1036, 789, 741, 467, 448. ^1^H NMR (DMSO-*d*6) δ 7.54 (m, 6H), 6.87 (s, 1H), 3.90 (m, 4H), 2.21 (s, 3H), 1.40 (t, *J* = 7.2 Hz, 3H), 1.33 (t, *J* = 7.2 Hz, 3H). ^13^C NMR (DMSO-*d*6) δ 185.1, 148.4, 136.2, 135.7, 135.0, 129.7, 129.6, 129.5, 128.2, 127.8, 125.0, 118.3, 114.3, 74.7, 54.7, 54.6, 15.6, 10.8, 10.6. MS (EI) *m*/*z*: 480.1 (M^+^-BF_4,_ 11%) for C_21_H_20_^79^BrClNOS_2_]^+^.

#### 3.1.7. 2-*N*,*N*-Diethylamino-6,8-dibromo-4-(4-chlorophenyl)-7-methyl-4*H*-1,3-dithiol[4,5-*c*]chromen-2-ylium tetrafluoroborate (**4d**)

Colorless crystals (0.51 g, 79%). M.p. 241–242 °C. IR (ATR, cm^−1^)1561, 1450, 1228, 1154, 1041, 717, 489. ^1^H NMR (DMSO-*d*6) δ 7.81 (s, 1H), 7.51 (m, 4H), 6.99 (s, 1H), 3.92 (m, 4H), 2.19 (s, 3H), 1.41 (t, *J* = 7.2 Hz, 3H), 1.34 (t, *J* = 7.2 Hz, 3H). ^13^C NMR (DMSO-*d*6) δ 185.1, 147.5, 140.7, 135.7, 135.2, 129.8, 129.6, 128.3, 127.2, 127.1, 117.4, 117.1, 114.8, 75.5, 54.8, 54.6, 24.3, 10.8, 10.6. MS (EI) *m*/*z*: 557.9 (M^+^-BF_4,_ 8%) for C_21_H_19_^79^Br_2_ClNOS_2_]^+^.

#### 3.1.8. 2-*N*,*N*-Diethylamino-4-(4-chlorophenyl)-6,8-diiodo-4*H*-1,3-dithiol[4,5-*c*]chromen-2-ylium tetrafluoroborate (**4e**)

Colorless crystals (0.54 g, 75%). M.p. 240–241 °C. IR (ATR, cm^−1^) 1596, 1544, 1427, 1224, 1076, 990, 809, 701, 620. ^1^H NMR (DMSO-*d*6) δ 8.13 (d, *J* = 1.9 Hz, 1H), 7.83 (d, *J* = 1.9 Hz, 1H), 7.55 (m, 4H), 7.00 (s, 1H), 3.92 (m, 4H), 1.43 (t, *J* = 6.9 Hz, 3H), 1.33 (t, *J* = 6.9 Hz, 3H). ^13^C NMR (DMSO-*d*6) δ 184.5, 149.8, 147.3, 135.1, 134.6, 132.6, 129.3, 129.1, 128.3, 126.6, 118.4, 88.1, 87.3, 74.9, 54.2, 54.1, 10.3, 10.0. MS (EI) *m*/*z*: 639.8 (M^+^-BF_4,_ 14%) for C_20_H_17_ClI_2_NOS_2_]^+^.

### 3.2. X-Ray Structure Determination

X-ray diffraction data were collected with an Oxford-Diffraction XCALIBUR E CCD diffractometer equipped with graphite-monochromated MoK radiation. A single crystal was positioned at 40 mm from the detector, and 2552 frames were measured each at 15 and 100 s over a 1° scan. The unit cell determination and data integration were carried out using the CrysAlis package of Oxford Diffraction [26]. The structures were solved by direct methods using Olex2 [27] software with the SHELXS structure solution program and refined by full-matrix least-squares on F^2^ with SHELXL-2015 [28]. Hydrogen atoms attached to carbon were placed in fixed, idealized positions and refined as rigidly bonded to the corresponding non-hydrogen atoms. Whenever necessary, restraints were imposed on geometry and displacement parameters of disordered fragments. CCDC-1919638 contains the supplementary crystallographic data for this contribution. These data can be obtained free of charge via www.ccdc.cam.ac.uk/conts/retrieving.html (or from the Cambridge Crystallographic Data Centre, 12 Union Road, Cambridge CB2 1EZ, UK; fax: (+44) 1223-336-033; or deposit@ccdc.ca.ac.uk).

#### Crystal Data for Compound **4b**

C_21_H_20_BrClF_4_NOS_2_ (Mr = 568.67 g⋅mol-1), monoclinic, *a* = 8.9673(6) Å, *b* = 14.7904(8) Å, *c* = 17.5523(9), *β* = 92.335(5)°, *V* = 2326.0(2) Å^3^, T = 180 K, space group *P*2_1_/c, Z = 4, 10,919 coll. refl., 4114 indep. (*R*_int_ = 0.0608), Gof = 1.027, *R*_1_ = 0.0661, w*R*(*F*^2^) = 0.1447. CCDC–1919638.

## 4. Materials and Methods

### 4.1. General Information

Human osteosarcoma (HOS) and MCF7 cell lines were purchased from CLS Cell Lines Service GmbH (Eppelheim, Germany) and Normal Human Dermal Fibroblasts (NHDF) cell line from PromoCell, (Heidelberg, Germany); Eagle’s Minimal Essential Medium alpha (aMEM), Dulbecco’s Modified Eagle Medium (DMEM) without phenol red, 1% Penicillin–Streptomycin–Amphotericin B mixture (10K/10K/25 µg in 100 mL) and Trypsin–Versene (EDTA) mixture from Lonza, Verviers, Begium; fetal bovine serum (FBS) from Biochrom GmbH, Germany, CellTiter 96^®^ Aqueous One Solution Cell Proliferation Assay from Promega (Madison, WI, USA); Tryple from Gibco, (Langley, VA, USA); LIVE/DEAD Viability/Cytotoxicity Kit and phosphate buffered saline (PBS) from Invitrogen, (Eugene, OR, USA), 35-mm Cell Imaging Dishes from Eppendorf (Hamburg, Germany).

Absorbance at 490 nm from MTS assay was measured with an iMark plate reader (Biorad, Hercules, CA, USA) and images for live/dead staining and wound scratch assays were recorded with a Leica DMI 3000B inverted microscope (Wetzlar, Germany).

### 4.2. Cell Culture

All cell lines were cultured in alpha-MEM medium supplemented with 10% FBS and 1% penicillin–streptomycin–amphotericin B mixture under 5% CO_2_ humidified atmosphere at 37 °C. For passaging cells, Trypsin–Versene was used for HOS and MCF7 cells while Tryple was used for NHDF cells.

Stock solutions in DMSO were prepared for each compound to be tested and working solutions were prepared by diluting the stock solutions 100 times with cell culture medium. Negative control cells were treated with cell culture medium containing 1% DMSO.

### 4.3. Wound Scratch Assay

A stock solution of 500 µg/mL in DMSO was prepared for each compound and working solutions of 5 µg/mL were prepared by diluting the stock solution with cell culture medium.

For wound scratch assay, cells were seeded in 24-well plates at a density of 24 × 10^4^ cells/well for HOS cell line and 12 × 10^4^ cells/well for MCF7 cell line. After 48 h a wound was performed by scratching the cell culture in the form of a cross in the center of the well with a pipette tip. Wells were washed with PBS and cells were treated with the compounds to be tested, using 5 µg/mL solutions in cell culture medium except control wells which were treated with 1% DMSO in cell culture medium. Pictures with cell cultures were acquired under microscope and these were denoted as time 0 in the experiment. After 4 h, the compounds solutions were removed and replaced with fresh culture medium. The following day, cell cultures were photographed again marking the 24 h time point in the experiment. Pictures were analyzed using TScratch version 1.0 software developed by the Koumoutsakos group (CSE Lab), at ETH Zürich [29]. First, open wound area is calculated as a percentage of open image area at 24 h compared to time 0 of the experiment: (open wound area 24 h) = 100 x (open image area 24 h)/(open image area 0 h). Finally a graph is made representing the ratio between open wound areas of treated samples and control.

### 4.4. MTS Assay

For MTS assay, cells were plated on 96-well plates at densities of 10 × 10^3^, 7 × 10^3^ and 5 × 10^3^ cells/well for HOS, MCF7 and NHDF cell lines respectively in 100 µL/well aMEM medium and incubated for 24 h. The next day the media were replaced with **4a** solutions in serial dilutions and the plates were incubated for another 48 h. The media containing the test compound were then replaced with 100 µL/well DMEM without phenol red and 20 µL/well MTS reagent and the plates were returned to the incubator for another 3 h. Following the last incubation, the plates were removed from the incubator and absorbance at 490 nm was recorded with a plate reader.

The relative cell viability is expressed as percentage of the viability of control (cells treated only with cell culture medium and 1% DMSO). Cumulated replicates from three independent experiments were used in the analysis.

### 4.5. Live/Dead Staining

A stock solution of 500 µg/mL in DMSO was prepared for **4a** and working solution of 5 µg/mL were prepared by diluting stock with cell culture medium.

Cells were seeded on 35-mm Cell Imaging Dishes with glass bottoms at a density of 24 × 10^4^, 12 × 10^4^ and 5 × 10^4^ cells/dish for HOS, MCF7 and NHDF cell lines respectively. The following day they were treated with 1 mL of 5 µg/mL solution of **4a** and incubated for 24 h. Finally, the **4a** treatment solution was replaced with 500 µL 50:50 mixture of LIVE/DEAD staining solution with DMEM without phenol red. Images were acquired after 15 min of staining and processed with ImageJ Fiji distribution version 1.52.

### 4.6. Data Analysis

Data analysis was performed with GraphPad Prism software version 7.00 for Windows (GraphPad Software, San Diego, CA). Results are presented as means ± standard error of the mean (S.E.M.). To determine statistical significance between two groups parametric unpaired *t*-test with Welch’s correction was applied. Where groups failed Shapiro–Wilk normality test, nonparametric unpaired Mann–Whitney test was used instead. The difference was considered significant when *p* < 0.05.

## 5. Conclusions

Four new tricyclic flavonoids have been synthesized and characterized. These, along a previously studied flavonoid, were tested against HOS and MCF7 human cancer lines using a wound scratch assay. The results showed that all five compounds inhibit cell proliferation, with derivatives **4a**–**c** being the most active. Further cytotoxicity tests suggested that flavonoid **4a** is less toxic for normal human cells when compared to cancerous ones, although more research is required to optimize this property. When referring to the antibacterial properties of **4a**, the previously reported minimum inhibitory concentration against *S. aureus* of 0.24 µg/mL is well under the concentration of 1 µg/mL, at which the tested compound begins to be too toxic for the NHDF cell line.

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
