# Peer review of "The Cytotoxic Properties of Some Tricyclic 1,3-Dithiolium Flavonoids"

_molecules, 2019, doi:10.3390/molecules24132459_

Round 1
Reviewer 1 Report
The work entitled "The Cytotoxic Properties of Some Tricyclic 1,3-Dithiolium Flavonoids" is interesting as there is a need to evaluate new possibilities of medicine.
The results show analyzes using 3 cell lines NHDF (normal fibroblasts), HOS (osteosarcoma) and MCF7 (breast cancer).
Some comments and suggestions:
Tables must be presented without being closed, because the closed format is used for frames. It is necessary review table 1
Perhaps in Wound scratch assay relative to the analyzes, it would be interesting use the relative values in the graph, as it seems to me that MCF7 cells do not show the same growth as HOS to facilitating, in this way, the visualization of the results
In line 138 - replaces Figure 44, by Figure 4.
In the discussion iten - To discuss better the importance of conducting cell viability tests, therefore, there is a need for a higher concentration of the material for anticancer activity.
Regarding the conclusion there is a need for better elaboration, to make it clear if the data of the previous article mentioned that the MIC was around 0.24 μg/ml, but in this work the concentrations necessary for cellular inhibition occurs around 5ug/ml.
Reviewer 2 Report
The paper presents research on the synthesis and evaluation of the biological activity of five flavonoid derivatives. The manuscript is potentially interesting for readers, but some modifications and answer for questions are necessary.The main weakness of the presented work is a small number of compounds and some assays limited to only one compound.
- It has not been explained what were the reasons for using such substituents? Why was the methyl group and iodine atoms (4d) included in the compounds? In discussion, some comments about structure-activity relationships should be added.
- It is not clear to me why only one compound 4a was selected for further research, if similar activity was also shown by 4b and 4c (Fig.3)
- Why were these cancer cell lines: HOS and MCF7 selected for testing anticancer properties?
- line 80: here is written about the structure of flavonoids, while the crystalline structure of only one compound has been described. Have you tried to obtain monocrystals for other compounds?
- In Fig. 4 on the x-axis one should keep the scale proportional to the value even if it is a column chart or replace it with a scatter plot (XY); not line plot.
The authors statistically compare the % viability of cells at each compound concentration, while the discussion of the relationship between viability and concentration (as a whole profile) for any (cancerous or normal) cell model would also be interesting. In addition to p-value and determination of 'statistical significance', it would be beneficial to calculate of 'effect size' as a way of quantifying the size of the difference between two groups.
- The discussion is somewhat inconsistent. Only compound 4a was selected for further testing and it was found that it is unfortunately also toxic for normal cells. In that case, why the other compounds have not been studied?
- No information whether and how the assumptions of parametric tests were checked (normality of distribution, homogeneity of variance).
Typographic errors:
Fig. 3a. the violet bar should be denoted as 4b, and the green one as 4c.
Scheme 2: typo
